# Heterogenous Nuclear Ribonucleoprotein H1 Promotes Colorectal Cancer Progression through the Stabilization of mRNA of Sphingosine-1-Phosphate Lyase 1

**DOI:** 10.3390/ijms21124514

**Published:** 2020-06-25

**Authors:** Keitaro Takahashi, Mikihiro Fujiya, Hiroaki Konishi, Yuki Murakami, Takuya Iwama, Takahiro Sasaki, Takehito Kunogi, Aki Sakatani, Katsuyoshi Ando, Nobuhiro Ueno, Shin Kashima, Kentaro Moriichi, Hiroki Tanabe, Toshikatsu Okumura

**Affiliations:** 1Division of Gastroenterology and Hematology/Oncology, Department of Medicine, Asahikawa Medical University, 2-1 Midorigaoka-higashi, Asahikawa, Hokkaido 078-8510, Japan; ktakaha@asahikawa-med.ac.jp (K.T.); yuuki1228@asahikawa-med.ac.jp (Y.M.); ganmatakuya@asahikawa-med.ac.jp (T.I.); taka-sas@asahikawa-med.ac.jp (T.S.); kunogi@asahikawa-med.ac.jp (T.K.); sakatani@asahiawa-med.ac.jp (A.S.); k-ando@asahikawa-med.ac.jp (K.A.); u-eno@asahikawa-med.ac.jp (N.U.); shin1014@asahikawa-med.ac.jp (S.K.); morimori@asahikawa-med.ac.jp (K.M.); tant@asahikawa-med.ac.jp (H.T.); okumurat@asahikawa-med.ac.jp (T.O.); 2Department of Gastroenterology and Advanced Medical Sciences, Asahikawa Medical University, 2-1 Midorigaoka-higashi, Asahikawa, Hokkaido 078-8510, Japan; hkonishi@asahikawa-med.ac.jp; 3Department of Medicine, Knapp Center for Biomedical Discovery, The University of Chicago, 900 East 57th Street, 9th floor, Chicago, IL 60637, USA

**Keywords:** hnRNP H1, SGPL1, S1P, RNA-binding protein, colorectal cancer

## Abstract

The oncogenic properties of heterogeneous nuclear ribonucleoprotein H1 (hnRNP H1) have been reported, although the tumor-promoting mechanism remains unclear. We herein report the mechanism underlying colorectal cancer cell progression mediated by hnRNP H1. The growth of colorectal cancer cells was suppressed by hnRNP H1 downregulation. A terminal deoxynucleotidyl transferase dUTP nick-end labeling assay revealed the anti-apoptotic effect of hnRNP H1 in colorectal cancer cells. An RNA immunoprecipitation assay revealed that hnRNP H1 bound to sphingosine-1-phosphate lyase 1 (SGPL1). Reverse transcription-polymerase chain reaction revealed the high expression of hnRNP H1 mRNA in colorectal cancer cells and Spearman’s rank correlation coefficient showed a strong positive correlation between hnRNP H1 mRNA and SGPL1 mRNA. An siRNA of hnRNP H1 decreased SGPL1 mRNA expression in colorectal cancer cells, but not in non-tumorous cells. These findings suggested that hnRNP H1 increased SGPL1 mRNA expression specifically in cancer cells through direct binding. Targeted knockdown of hnRNP H1 or SGPL1 with siRNAs upregulated p53 phosphorylation and p53-associated molecules, resulting in cell growth inhibition, while hnRNP H1 upregulated the mRNA of SGPL1 and inhibited p53 activation, thereby promoting tumor cell growth. This is a novel mechanism underlying colorectal cancer cell progression mediated by hnRNP H1–SGPL1 mRNA stabilization.

## 1. Introduction

Colorectal cancer is one of the most frequent causes of cancer-related death worldwide [1]. The outcome of colorectal cancer has improved over the past several decades due to the development of new chemotherapy regimens combining cytotoxic drugs with molecular-targeted therapy. However, the mortality of advanced colorectal cancer remains high, underscoring the need for novel therapeutic targets [2].

Heterogeneous nuclear ribonucleoproteins (hnRNPs) are groups of RNA-binding proteins that have RNA recognition motifs in a domain sequence and contribute to multiple aspects of nucleic acid metabolism [3,4]. The hnRNPs have essential properties related to the maturation, transportation, stabilization and translation of RNAs and work as a key regulator of the development and differentiation of tissues and organs in mammalian cells [4]. Previous studies have shown that several hnRNPs have oncogenic properties, such as the induction of excess cell growth and exertion of an anti-apoptotic effect [5], but each hnRNP targets different RNAs depending on the cancer type or origin. Therefore, in order to determine the function of each hnRNP, the mechanisms of each hnRNP must be investigated in each cancer type and origin.

It has been reported that hnRNP H1, which belongs to the hnRNP family, is highly expressed in several cancers, including colorectal, head and neck, hepatocellular, pancreatic and laryngeal carcinomas [3,6,7]. In colorectal cancer cells, a study proposed that hnRNP H1 increased the mRNA and protein expression of a-raf, thereby preventing apoptosis [6]. That study speculated that hnRNP H1 was involved in the splicing of a-raf mRNA, while direct binding between hnRNP H1 and a-raf mRNA was not demonstrated. The roles of other target RNAs of hnRNP H1 have been unclear, although hnRNP H1 probably binds to multiple RNAs in colorectal cancer.

In the present study, we confirmed the tumor-promoting effect of hnRNP H1 and selected 10,914 mRNAs that bind to hnRNP H1 based on an immunoprecipitation-transcriptome assay in colorectal cancer cells. Our functional assay using an siRNA library identified sphingosine-1-phosphate lyase 1 (SGPL1), which is associated with sphingosine-1-phosphate (S1P) degradation, as the target of hnRNP H1 for mediating the tumor-promoting effects. Finally, we found that the p53-related genes cyclin G2 and CDKN1A were dysregulated in SGPL1-downregulated cells, illustrating a novel tumor-promoting mechanism involving hnRNP H1-upregulated SGPL1 mRNA.

## 2. Results

### 2.1. hnRNP H1 Promoted Cell Progression by Inhibiting Apoptosis in Colorectal Cancer Cells

To assess the tumor-promoting properties of hnRNP H1, an siRNA of hnRNP H1 #1 was transfected into HCT116, SW480 and SK-CO-1 cells. Cell growth was significantly reduced in HCT116 and SW480 cells transfected with the siRNA of hnRNP H1 #1 compared with cells transfected with scrambled RNA. The knockdown efficacies by reverse transcription (RT)-polymerase chain reaction (PCR) and western blotting are also shown in Appendix A. In contrast, cell growth was suppressed to a lesser degree or not at all by the downregulation of hnRNP H1 in HCEC-1CT or SK-CO-1 cells compared with HCT116 and SW480 cells (Figure 1A). To exclude the off-target effect of the siRNA of hnRNP H1 #1, siRNAs of hnRNP H1 #2 and #3 were transfected into HCT116 cells. Cell growth was significantly reduced in HCT116 cells transfected with the siRNAs of hnRNP H1 #2 and #3 compared with cells transfected with scrambled RNA (Figure 1B). The knockdown efficacy is shown in Appendix A. These data suggested that cell growth in non-tumorous cells and some cancer cells was not markedly influenced by hnRNP H1 expression.

To assess the tumor-promoting effect of hnRNP H1 in vivo, HCT116 cells were transplanted into nude mice, and the siRNA of hnRNP H1 #1 or scrambled RNA was directly injected into the tumor every day. Tumor growth was significantly reduced by treatment with the siRNA of hnRNP H1 #1 (Figure 1C).

To investigate the anti-apoptotic properties of hnRNP H1 in colorectal cancer cells, staining with the terminal deoxynucleotidyl transferase dUTP nick-end labeling (TUNEL) procedure was performed in HCT116 cells (Figure 1D). The number of TUNEL-positive cells were significantly increased in HCT116 cells transfected with the siRNA of hnRNP H1 #1, suggesting that hnRNP H1 inhibited the apoptosis of colorectal cancer cells.

These data confirmed that hnRNP H1 exerted tumor-promoting effects by inhibiting tumor cell apoptosis in colorectal cancer cells.

### 2.2. hnRNP H1 Was Highly Expressed in Colorectal Cancer Cells

To investigate the dysregulation of hnRNP H1 in colorectal cells, the expression of hnRNP H1 was compared between tumorous and non-tumorous cells. An analysis using the UCSC Xena system (https://xena.ucsc.edu/compare-tissue/), which included 286 colorectal cancer tissues and 41 normal colon tissues, revealed the overexpression of hnRNP H1 in colorectal cancer tissues (Figure 2A). RT-PCR using our institutional samples, which were collected from 28 normal colon tissues and 32 colorectal cancer tissues, showed the overexpression of hnRNP H1 mRNA in specimens of human cancerous lesions compared with non-cancerous lesions (Figure 2B). Western blotting showed the overexpression of hnRNP H1 in azoxymethane (AOM)/dextran sodium sulfate (DSS) carcinogenesis model mice compared with control mice (Figure 2C). These findings suggested that hnRNP H1 was overexpressed in colorectal cancer.

### 2.3. hnRNP H1 Upregulated SGPL1 mRNA and Promoted Colorectal Cancer Progression

To identify hnRNP H1-interacting mRNAs, cell lysate of HCT116 cells was immunoprecipitated using a hnRNP H1 antibody and RNA was extracted from the precipitate. A transcriptome analysis using RNA extracted from the precipitate revealed that 10,914 mRNAs directly bound to hnRNP H1 (value of fold change > 2, *p* < 0.05) (Appendix A). To identify mRNAs with hnRNP H1-regulated expression, a transcriptome analysis was performed in hnRNP H1-downregulated cells. This showed that the expression of 889 mRNAs was significantly changed in hnRNP H1-downregulated cells (absolute value of fold change > 2, *p* < 0.05) (Appendix A). These findings along with those from the immunoprecipitation (IP)-transcriptome assay using the hnRNP H1 antibody and the transcriptome analysis in hnRNP H1-downregulated cells suggested that the expression of 591 mRNAs was directly regulated by hnRNP H1 (Appendix A).

Of these 591 mRNAs, 54 apoptosis-related mRNAs were selected because hnRNP H1 regulated apoptosis in colorectal cancer cells (Table 1). To assess the tumor-promoting function of hnRNP H1-binding mRNAs, these 54 mRNAs were knocked down using the siRNA of each mRNA. SGPL1 downregulation showed the strongest inhibition of cell growth in HCT116 cells (Figure 3A). RNA immunoprecipitation combined with RT-PCR confirmed the direct binding of hnRNP H1 and SGPL1 mRNA (Figure 3B).

To confirm the influence of SGPL1 mRNA by hnRNP H1, the expression of SGPL1 mRNA was assessed in hnRNP H1-downregulated HCT116 cells. RT-PCR and western blotting showed a decrease in SGPL1 mRNA and protein expression in hnRNP H1-downregulated HCT116 cells (Figure 3C). Conversely, the downregulation of SGPL1 mRNA and protein was not observed in hnRNP H1-downregulated HECE-1CT cells (Figure 3D), suggesting that hnRNP H1 did not influence the expression of SGPL1 mRNA and protein in non-tumorous cells. To clarify the relationship between hnRNP H1–SGPL1 mRNA interaction and the tumor-suppressive effect, SGPL1 was knocked down in hnRNP H1-downregulated HCT116 cells. An additional tumor-suppressive effect was not observed by SGPL1 downregulation (Figure 3E). In addition, to examine whether or not the ectopic expression of SGPL1 rescues hnRNP H1 knockdown-mediated reduction of cell viability, S1P was added to hnRNP H1-downregulated HCT116 cells. Cell growth was significantly increased in hnRNP H1-downregulated HCT116 cells with S1P compared to those without S1P (Figure 3F).

These data suggested that hnRNP H1 promoted the growth of colorectal cancer cells, but not non-cancerous cells, through the upregulation of SGPL1 mRNA.

### 2.4. SGPL1 mRNA Had Correlation with the Expression of hnRNP H1 mRNA in Colorectal Cancer Cells

An RT-PCR analysis of the surgically resected samples in our institution showed that SGPL1 mRNA tended to be highly expressed in colorectal cancer tissues (*n* = 32) compared with normal mucosa tissues (*n* = 28) (*p*-Value = 0.10, Figure 4A). Spearman’s rank correlation coefficient revealed strong positive correlation (ρ = 0.736, *p*-Value < 0.05) between hnRNP H1 mRNA and SGPL1 mRNA (Figure 4B).

### 2.5. The Intracellular S1P Expression Was not Changed in hnRNP H1- or SGPL1-Downregulated HCT116 Cells

Previous investigations indicated that SGPL1 degraded the sphingolipid component S1P to phosphoethanolamine and hexadecenal, and inhibited the proliferation of epithelial cells [8]. To clarify whether or not upregulated SGPL1 influenced S1P degradation in colorectal cancer cells, the expression of S1P was assessed in hnRNP H1- or SGPL1-downregulated HCT116 cells by an enzyme-linked immunosorbent assay (ELISA). The intracellular S1P expression was not changed in hnRNP H1- or SGPL1-downregulated HCT116 cells (Figure 5), suggesting that upregulated SGPL1 did not influence sphingolipid metabolism in colorectal cancer cells.

### 2.6. p53 Activation Is a Key Regulator of Tumor-Promoting Mechanisms Mediating hnRNP H1–SGPL1 mRNA Stabilization

To clarify the mechanism underlying the tumor-suppressive effect of hnRNP H1-upregulated SGPL1, a cDNA array analysis was performed in SGPL1-downregulated cells. A total of 166 mRNAs were changed, with an absolute value > 2-fold change, and the levels of two cell cycle-related genes, CDKN1A and cyclin G2, were increased in SGPL1-downregulated HCT116 cells (Table 2). Previous investigations showed that p53 is the main regulator of the expression of CDKN1A and cyclin G2 [9,10]. In addition, SGPL1 is known to be associated with cell apoptosis through p53 activation [11]. Western blotting revealed that p53 was significantly activated in SGPL1- as well as hnRNP H1-downregulated cells, whereas the amount of p53 did not change (Figure 6A). The tumor-suppressive effect induced by downregulating hnRNP H1 or SGPL1 was diminished by treatment with the p53 inhibitor, pifithrin-μ (Figure 6B). In addition, the tumor-suppressive effect induced by downregulating hnRNP H1 or SGPL1 was less influential in HCT116 p53^−/−^ cells than in HCT116 parental cells (Figure 6C; knockdown efficacy shown in Appendix A).

These findings suggested that hnRNP H1-upregulated SGPL1 inhibited p53 phosphorylation and thus promoted colorectal cancer cell progression. 

## 3. Discussion

The present study showed that hnRNP H1 promoted colorectal cancer cell progression by inhibiting cell apoptosis and by binding to and stabilizing SGPL1 mRNA in colorectal cancer cells. The downregulation of hnRNP H1 or SGPL1 dramatically reduced the growth of colorectal cancer in vitro and in vivo, showing SGPL1 to be a novel binding partner and mediator of the tumor-promoting effect of hnRNP H1. Notably, we found that hnRNP H1 downregulation did not influence the expression of SGPL1 mRNA in non-tumorous cells, suggesting that hnRNP H1-induced upregulation of SGPL1 mRNA was cancer-specific and might be a new target of cancer therapy.

SGPL1 is a membrane-bound enzyme that irreversibly degrades S1P by cleaving it into hexadecenal and phosphoethanolamine [8,12,13]. Sphingolipid metabolites, including ceramide, sphingosine and S1P, have emerged as bioactive signaling molecules that regulate cell movement, differentiation, survival, inflammation, angiogenesis, calcium homeostasis and immunity [14]. Ceramide is considered to be responsible for antiproliferative responses, such as the inhibition of cell growth and induction of apoptosis, while S1P has opposite functions, including the promotion of cancer cell growth and angiogenesis, based on the sphingosine rheostat or ceramide-S1P rheostat theory [8,12,15,16]. In line with the sphingolipid rheostat theory, previous reports showed that SGPL1 was downregulated in colon cancer, leading to S1P accumulation in neoplastic intestinal tissues [11]. The SGPL1 expression and activity were also significantly reduced in adenomas and colitis-associated cancer in the mouse model compared to the control model [11,17]. These findings suggested that SGPL1 might serve an anti-oncogenic role in mice models [8]. However, in contrast to these previous findings, the upregulation of SGPL1 and no increase in S1P levels were recently reported in human colon cancer tissues [18]. In addition, there has been no report describing an increase in S1P levels in human colon cancer tissues. Our results also showed that S1P expression was not changed, whereas the S1P-cleaved enzyme SGPL1 was upregulated, in colorectal cancer cells. These data highlight the novel role of SGPL1 in tumor progression that differs from its role in sphingolipid metabolism.

We comprehensively analyzed the changes in mRNA expression in SGPL1-downregulated cells using a transcriptome analysis and identified 166 mRNAs with more than twofold changes in their expressions. Of these mRNAs, CDKN1A and cyclin G2 were identified as apoptosis-associated mRNAs with expressions regulated by phosphorylated p53 [9,10]. Subsequently, we found that phosphorylated p53 was significantly increased in hnRNP H1- and SGPL1-downregulated cells. In fact, the growth of HCT116 and SW480 cells was significantly decreased by the inhibition of hnRNP H1, suggesting that hnRNP H1 played a pivotal role in the growth of these two cell lines. In contrast, the growth of SK-CO-1 was not changed when hnRNP H1 was knocked down. It has been reported that the growth of SK-CO-1 does not depend on the p53 pathway [19], which is closely influenced by hnRNP H1 knockdown, thus suggesting that hnRNP H1 knockdown inhibited the growth of cells depending on the p53 pathway. Taken together, these findings suggest that the hnRNP H1-induced stabilization of SGPL1 inhibited p53 activation, leading to the dysregulation of cell cycle and apoptosis in colorectal cancer cells.

In the present study, HCT116 p53^−/−^ cells were used to validate the knockdown effect of hnRNP H1. The growth-suppressing effect of the downregulation of hnRNP H1 or SGPL1 was diminished in p53 null HCT116 cells. However, the suppressive effects were not completely repressed in p53^−/−^ cells, suggesting that other pathways were associated with cell growth promoted by hnRNP H1–SPGPL1. A previous study identified a-raf mRNA as a target of hnRNP H1 using a transcriptome analysis in hnRNP H1-downregulated cells [6]. The study showed that the expressions of a-raf mRNA and protein were increased in colorectal cancer DLD-1 cells, following the inactivation of the pro-apoptotic MST2 and then preventing apoptosis. The authors speculated that hnRNP H1 was involved in the splicing of a-raf mRNA because hnRNP H1 is a splice factor, which is required for the correct transcription and expression of a-raf. However, the direct binding between hnRNP H1 and a-raf mRNA was not demonstrated in this study. We also examined the expression of a-raf mRNA in hnRNP H1-downregulated cells and identified small, non-significant changes (approximately 1.5-fold reductions) in the mRNA (data not shown). Our RNA immunoprecipitation combined with RT-PCR assay confirmed the direct binding of hnRNP H1 and SGPL1 mRNA, and our transcriptome analysis in hnRNP H1-downregulated cells subsequently showed that hnRNP H1 contributed to the increase in SGPL1 expression. Finally, in SGPL1-downregulated cells, tumor cell apoptosis was induced, and the effect of hnRNP siRNA was negated, clearly indicating that hnRNP H1 function was mediated by SGPL1.

In conclusion, the present study demonstrated that hnRNP H1-induced stabilization of SGPL1 mRNA promoted tumor progression through the inhibition of p53 phosphorylation in colorectal cancer cells, but not in non-tumorous cells. This suggests that the hnRNP H1–SGPL1 mRNA interaction might be a novel therapeutic target of colorectal cancer treatment. Further analyses are needed to identify the mechanism underlying the cancer-specific hnRNP H1–SGPL1 mRNA interaction, such as pre- and/or post-transcriptional alterations of hnRNP H1 and/or SGPL1, and to identify other pathways associated with cell growth promoted by hnRNP H1–SPGPL1, including microRNAs and long-ncRNA pathways.

## 4. Materials and Methods

### 4.1. Cell Culture

The human colon cancer cell lines (HCT116 and SW480, purchased from ATCC; HCT116 parental and isogenic HCT116 p53^−/−^ sublines, purchased from Horizon) and the human colon epithelial cell line (HCEC-1CT, purchased from Evercyte) were grown in McCoy’s 5A medium (HCT116), Roswell Park Memorial Institute 1640 medium (SW480), high-glucose Dulbecco’s Modified Eagle’s medium (SK-CO-1) and ColoUp medium (HCEC-1CT) supplemented with 10% (*v*/*v*) fetal bovine serum (FBS), 2 mM of L-glutamine, 50 U/mL of penicillin and 50 µg/mL of streptomycin in a humidified atmosphere of 5% CO_2_. The cells were plated at a density of 10^5^ cells/cm^2^.

### 4.2. Human Colorectal Specimens of Colorectal Cancers and Normal Colons

Twenty-eight colorectal cancer patients from Asahikawa Medical University Hospital were enrolled in this study. After therapeutic surgical resection, 32 samples of colorectal cancer were collected from the cancerous lesions and 28 samples of normal colonic mucosa were collected from the non-tumorous lesions. RNA was collected from formalin-fixed paraffin-embedded (FFPE) samples of the 32 cancerous lesions and 28 non-cancerous lesions using the RecoverAll Total Nucleic Acid Isolation Kit for FFPE (Thermo Fisher Scientific K.K., MA, USA). Written informed consent was obtained from all patients and the ethics committee of Asahikawa Medical University gave its approval for this study.

### 4.3. Protein Extraction

The cells were lysed for western blotting and immunoprecipitation using the NP-40 cell lysis buffer (Thermo Fisher Scientific, MA, USA) containing RNasin (Promega, WI, USA) and a complete protease inhibitor cocktail (Roche Molecular Biochemicals, Indianapolis, IN, USA).

### 4.4. Western Blotting

Equal amounts of protein were resolved using sodium dodecyl sulfate-polyacrylamide gel electrophoresis (SDS-PAGE) (12.5%), blotted to a nitrocellulose membrane and then blocked in phosphate-buffered saline (PBS) with 0.05% (*v*/*v*) Tween-20 (T-PBS) containing 1% (*w*/*v*) bovine serum albumin. Each blot was incubated overnight at 4 °C with primary antibodies. Primary antibodies for hnRNP H1 (ab10374) and p53 (phospho S15) (ab1431) were purchased from Abcam, primary antibody for SGPL1 (AF5535) was purchased from R&D Systems (Minneapolis, MN, USA) and primary antibody for p53 (506135) was purchased from Calbiochem. The blots were then washed in T-PBS, incubated with horseradish peroxidase (HRP)-conjugated secondary antibodies (R&D Systems), washed in T-PBS and then developed using a Super-Signal West Pico enhanced chemiluminescence system (Thermo Fisher Scientific). The averaged protein expression was normalized to actin expression (BD Transduction Laboratories, Lexington, KY, USA).

### 4.5. Transcriptome Analyses

RNA was purified using a mirVana miRNA isolation kit (Thermo Fisher Scientific). RNA libraries were generated using an Ion Total RNA-Seq kit v2 (Life Technologies) according to the manufacturer’s instructions. The RNA libraries were then processed for emulsion PCR using an Ion OneTouch^TM^ system and an Ion OneTouch 200 Template kit v3 (Life Technologies). Template-positive Ion Sphere^TM^ particles were enriched and purified for the sequencing reaction with an Ion OneTouch^TM^ ES system (Life Technologies). The template-positive Ion Sphere^TM^ particles were then applied on Ion PI™ chips (Life Technologies) and a high-throughput sequencing reaction was carried out using an Ion Proton™ semiconductor sequencer (Life Technologies). All of the sequencing data were mapped on a human reference genome sequence (GRCh37/hg19) using the Torrent Suite software program (Life Technologies). The expression analysis for each sample was imported into the CLC Genomics Workbench software program (CLC Bio, Aarhus, Denmark) and the significance of the differences among the samples was determined by an unpaired *t*-test. The gene ontology (GO) analysis was performed using the MetaCore software.

### 4.6. Real-Time PCR

Total RNA was extracted using an RNeasy mini kit (Qiagen, Tokyo, Japan) according to the manufacturer’s instructions. The mRNAs were reverse transcribed using a high-capacity cDNA reverse transcription kit (Applied Biosystems, Foster City, CA, USA). Gene expression was measured in duplicate, using specific primers for hnRNP H1 (Hs04979572_g1) and SGPL1 (Hs00393705_m1), which were purchased from Applied Biosystems.

### 4.7. cDNA Analysis

The mRNA profiling was investigated using a Clariom S array (Thermo Fisher Scientific, MA, USA). A difference more than 2-fold was considered to indicate a significant change.

### 4.8. Binding Assay

The lysates were clarified using centrifugation for 10 min at 12,000 rpm and then immunoprecipitated using IgG or the hnRNP H1 antibody (1 μg each) with a Dynabeads immunoprecipitation kit (VERITAS Corporation, Tokyo, Japan). RNA was extracted from the beads using phenol–chloroform extraction and purified using the mirVana™ isolation kit (ThermoFisher Scientific). RT-PCR was then performed using this RNA sample and the spectrum data were acquired using an Applied Biosystems 7300 real-time PCR system.

### 4.9. SRB Assay

The cells were seeded on 96-well microplates at 1.0 × 10^4^ cells per well for 24 h before stimulation. The cells were fixed in 5% trichloroacetic acid (TCA) for 1 h at 4 °C and washed in distilled water for 4 times. The microplates were then dehydrated at room temperature, stained in 0.057% SRB solution, washed in 0.1% acetic acid and re-dehydrated. The stained cells were lysed in 10 mM Tris-buffer and the optical density (OD) was measured at 510 nm.

### 4.10. Staining with Terminal Deoxynucleotidyl Transferase (TdT)-Mediated dUTP Nick-End Labeling (TUNEL)

The cells were plated on chamber slides. The slides were fixed in 4% paraformaldehyde and washed extensively with PBS. The slides were stained using the In Situ Cell Death Detection Kit and TMR red (Roche Diagnostic, IN, USA) according to the manufacturer’s instructions. The cells were mounted with an anti-fade mounting medium and TUNEL-positive cells were visualized by fluorescence microscopy (KEYENCE Corporation, Osaka, Japan).

### 4.11. Xenografts

The protocols of the animal experiments were approved by the Asahikawa Medical University Institutional Animal Care and Use Committee (ethical approval no. 19027). HCT116 cells (2 × 10^6^ cells) were injected into male BALB/c nude mice. The siRNA or control RNA was transfected daily using a GENOMONE-Si transfection kit (Ishihara Sangyo, Co, Ltd., Osaka, Japan) into the transplanted tumor via local injection.

### 4.12. RNAs and Transfections

A negative control with a scrambled RNA sequence was prepared by annealing two synthetic RNAs as follows: sense, 5′-UACGUACUAUCGCGCGGAU-3′ and antisense, 5′-AUCCGCGCGAUAGUACGUA-3′. The sequence of the siRNA of hnRNP H1 #1 was CGACCAAGUUUUACAGGAA (dTdT) and UUCCUGUAAAACUUGGUCG (dTdT), purchased from Bioneer Inc. (Daejeon, Korea). The sequence of the siRNA of hnRNP H1 #2 was GGAGCUGGCUUUGAGAGGAUU and UCCUCUCAAAGCCAGCUCCUU and that of #3 was GAAUAGGGCACAGGUAUAUUU and AUAUACCUGUGCCCUAUUCUU, synthesized by Hokkaido System Science Co., Ltd. (Hokkaido, Japan). The sequence of the siRNA of SGPL1 was CUGUACUACUGACCCAACA (dTdT) and UGUUGGGUCAGUAGUACAG (dTdT), purchased from Bioneer Inc. The cells were seeded 24 h prior to transfection, and transfection was performed using the Lipofectamine RNAiMax reagent (Invitrogen, CA, USA).

### 4.13. ELISA

Cell lysates were prepared using an NP-40 cell lysis buffer containing a phosphatase inhibitor and a protease inhibitor. The contents of intracellular S1P were determined using an S1P ELISA kit (Echelon Biosciences Inc., UT, USA) according to manufacturer’s instructions.

### 4.14. Sphingosine-1-Phosphate (S1P)

S1P was purchased from Cayman Chemical (CAS registry No. 26993-30-6).

### 4.15. Statistical Analyses

The assay data were analyzed using Student’s *t*-test and an analysis of variance (ANOVA). The clinical sample data were analyzed using the Mann–Whitney U test and *p*-Values < 0.05 were considered to be statistically significant.

## Figures and Tables

**Figure 1 ijms-21-04514-f001:**
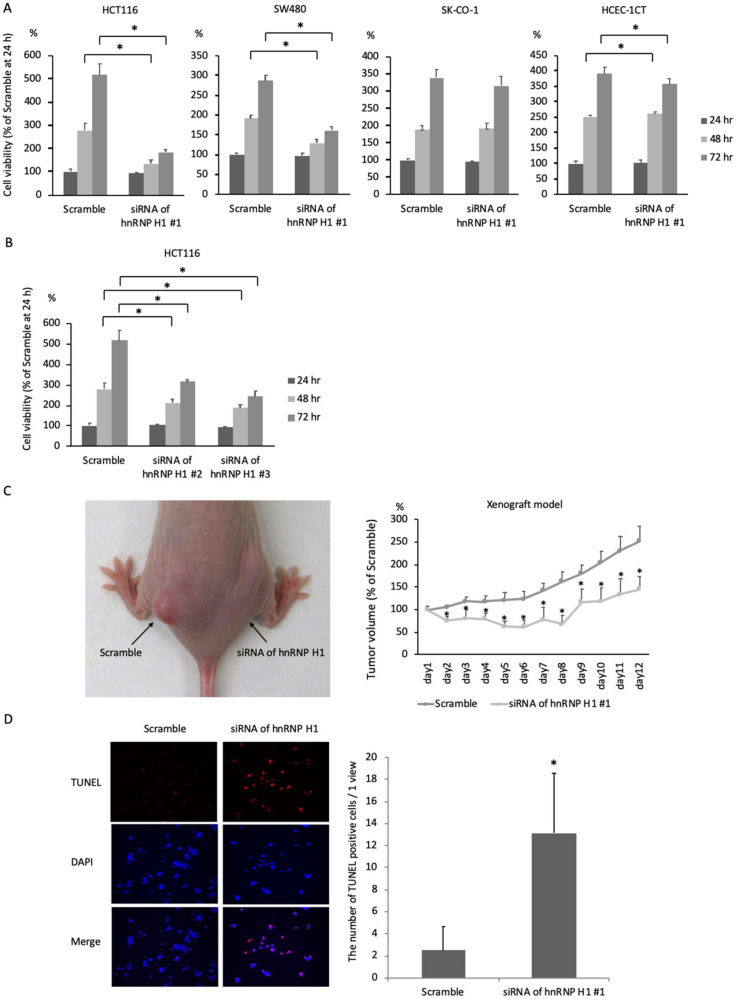
Heterogeneous nuclear ribonucleoprotein H1 (hnRNP H1) promoted cell progression by inhibiting apoptosis in colorectal cancer cells. (**A**) The sulforhodamine B (SRB) assay (*n* = 5) showed that cell growth was significantly reduced in HCT116 and SW480 cells transfected with an siRNA of hnRNP H1 compared with those transfected with scrambled RNA. The SRB assay (*n* = 5) showed that cell growth was suppressed to a lesser degree or not at all by the downregulation of hnRNP H1 in HCEC-1CT or SK-CO-1 cells. (**B**) The SRB assay (*n* = 5) showed that cell growth was significantly reduced in HCT116 cells transfected with siRNAs of hnRNP H1 #2 and #3 compared with cells transfected with scrambled RNA. (**C**) HCT116 cells were transplanted into nude mice, and the siRNA of hnRNP H1 #1 or scrambled RNA was injected daily. Tumor growth was significantly reduced by treatment with the siRNA of hnRNP H1 #1. (**D**) The number of terminal deoxynucleotidyl transferase dUTP nick-end labeling (TUNEL)-positive cells was significantly higher in HCT116 cells transfected with the siRNA of hnRNP H1 #1 than in those transfected with scrambled RNA. The error bars show the standard deviation (SD). * *p* < 0.05 by Student’s *t*-test.

**Figure 2 ijms-21-04514-f002:**
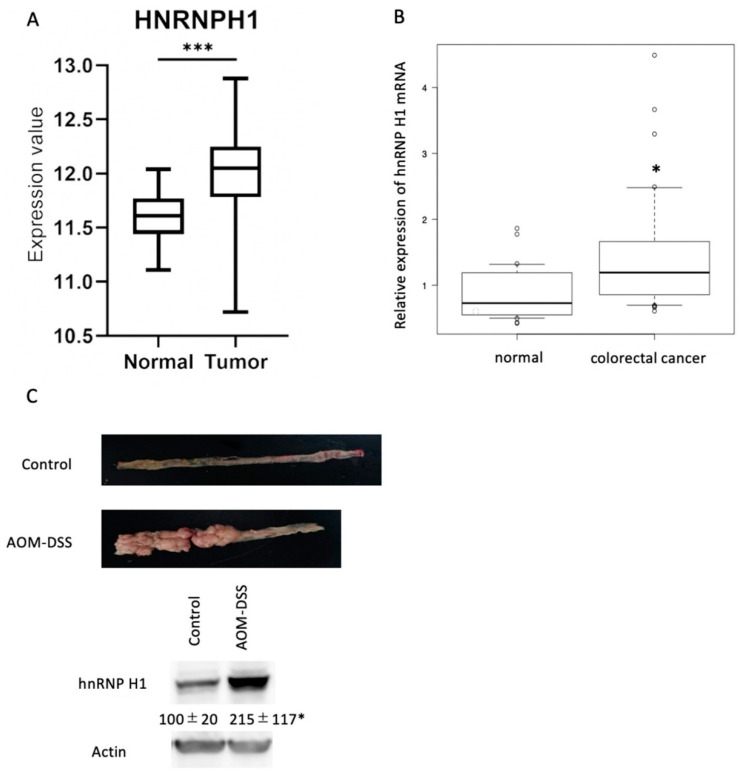
hnRNP H1 was highly induced in colorectal cancer cells. (**A**) An analysis using the UCSC Xena system revealed the overexpression of hnRNP H1 in colorectal cancer tissues (*n* = 286) compared with normal colon tissues (*n* = 41). (**B**) RT-PCR revealed the overexpression of hnRNP H1 mRNA in surgically removed specimens of human cancerous lesions (*n* = 32) compared with non-tumorous lesions (*n* = 28). (**C**) Western blotting showed the overexpression of hnRNP H1 in the colon of azoxymethane (AOM)/dextran sodium sulfate (DSS) carcinogenesis model mice compared with control mice. *** *p* < 0.0001, * *p* < 0.05 by Student’s *t*-test.

**Figure 3 ijms-21-04514-f003:**
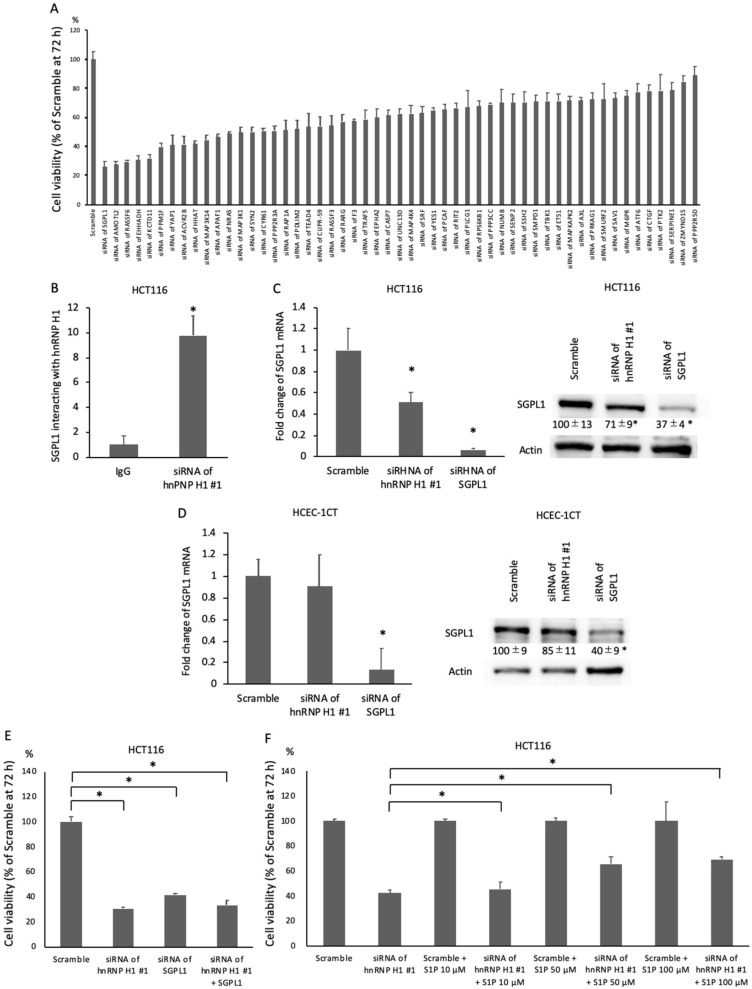
hnRNP H1 upregulated sphingosine-1-phosphate lyase 1 (SGPL1) mRNA and promoted colorectal cancer progression. (**A**) The SRB assay (*n* = 5) showed that SGPL1 downregulation resulted in the strongest inhibition of cell growth in HCT116 cells. (**B**) RNA immunoprecipitation combined with RT-PCR (*n* = 3) confirmed the direct binding of hnRNP H1 and SGPL1 mRNA. (**C**) RT-PCR (*n* = 3) and western blotting (*n* = 3) showed the decrease in SGPL1 mRNA and protein in hnRNP H1-downregulated HCT116 cells. (**D**) RT-PCR (*n* = 3) and western blotting (*n* = 3) showed that the downregulation of SGPL1 mRNA and protein was not observed in hnRNP H1-downregulated HECE-1CT cells. (**E**) The SRB assay (*n* = 5) showed that an additional tumor-suppressive effect was not observed by SGPL1 downregulation. (**F**) The SRB assay (*n* = 5) showed that cell growth was significantly increased in the hnRNP H1-downregulated HCT116 cells with S1P in a concentration-dependent manner compared to those without sphingosine-1-phosphate (S1P). The error bars show the SD. * *p* < 0.05 by Student’s *t*-test.

**Figure 4 ijms-21-04514-f004:**
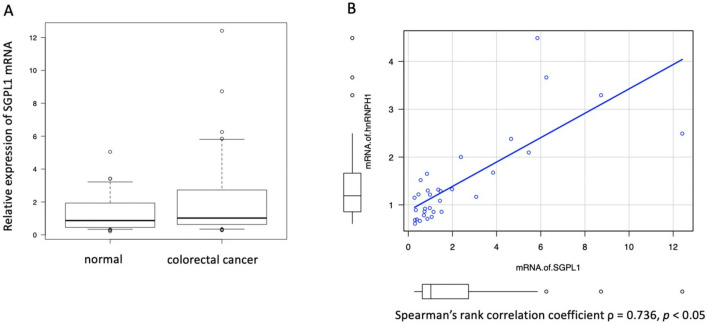
The expression of SGPL1 mRNA and hnRNP H1 mRNA had a strong positive correlation. RT-PCR showed that SGPL1 mRNA tended to be highly expressed in colorectal cancer tissues (*n* = 32) compared with normal mucosa tissues (*n* = 28, *p*-Value = 0.01) (**A**) Spearman’s rank correlation revealed a strong positive correlation coefficient between hnRNP H1 mRNA and SGPL1 mRNA. (**B**) **c* by Student’s *t*-test.

**Figure 5 ijms-21-04514-f005:**
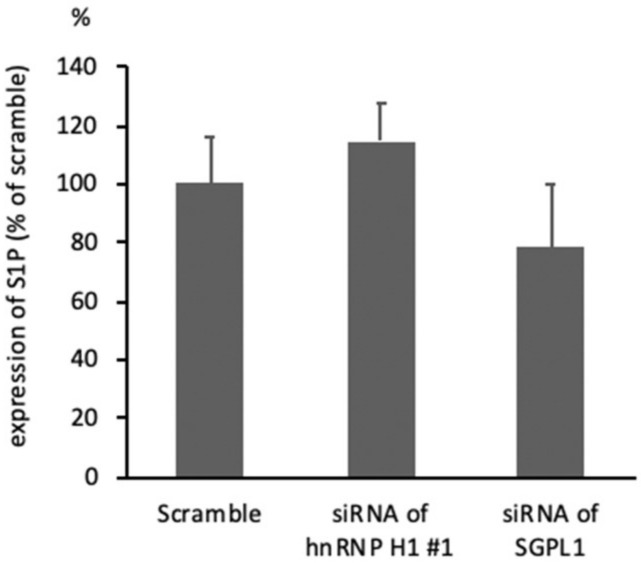
The expression of S1P in hnRNP H1- or SGPL1-downregulated HCT116 cells as assessed by ELISA. ELISA (*n* = 3) showed that the intracellular S1P expression was not changed in hnRNP H1- or SGPL1-downregulated HCT116 cells. The error bars show the SD.

**Figure 6 ijms-21-04514-f006:**
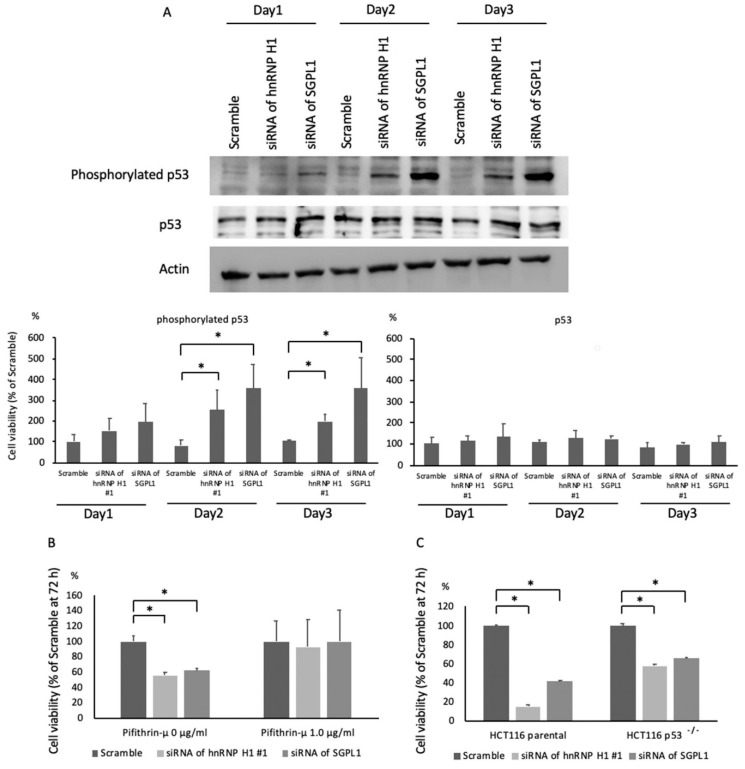
siRNA of hnRNP H1 or SGPL1 upregulated p53 phosphorylation and inhibited cell growth. (**A**) Western blotting (*n* = 3) revealed that phosphorylated p53 was significantly increased in hnRNP H1- and SGPL1-downregulated cells, whereas p53 did not change. (**B**) The SRB assay (*n* = 5) revealed that the tumor-suppressive effect induced by downregulating hnRNP H1 or SGPL1 after 72 h in cultured HCT116 cells was diminished by treatment with the p53 inhibitor, pifithrin-μ. (**C**) The SRB assay (*n* = 5) revealed that the tumor-suppressive effect induced by downregulating hnRNP H1 or SGPL1 was less influential in HCT116 p53^−/−^ cells than in HCT116 parental cells (**C**). The error bars show the SD. * *p* < 0.05 by Student’s *t*-test.

**Table 1 ijms-21-04514-t001:** List of apoptosis-related mRNAs regulated by hnRNP H1.

	IP-Transcriptome Analysis (hnRNP H1/IgG)		Transcriptome Analysis (siRNA of hnRNP H1/Scramble)			IP-Transcriptome Analysis (hnRNP H1/IgG)		Transcriptome Analysis (siRNA of hnRNP H1/Scramble)	
	Fold Change	*p*-Value	Fold Change	*p*-Value		Fold Change	*p*-Value	Fold Change	*p*-Value
ACVR2B	15.144	0.0006	−2.574	0.026	PPP2R3A	2.530	0.0060	−2.047	0.0200
AJUBA	11.234	0.0004	−2.678	0.039	PPP2R5D	2.298	0.0019	−2.490	0.0442
AMOTL2	2.834	0.0002	−2.737	0.019	PPP3CC	2.502	0.0176	−2.200	0.0350
APAF1	10.245	0.0002	−2.008	0.037	PRKAG1	12.766	0.0000	−3.006	0.0166
ATF6	6.076	0.0014	−2.254	0.044	PTK2	2.441	0.0002	−2.921	0.0063
AXL	2.068	0.0030	−2.603	0.018	RAP1A	35.974	0.0001	−2.280	0.0242
CASP7	6.792	0.0007	−4.722	0.001	RARG	3.885	0.0000	−2.098	0.0291
CLIP3	15.400	0.0225	−2.544	0.020	RASSF3	17.411	0.0002	−2.738	0.0104
CTGF	4.658	0.0156	−4.238	0.009	RASSF6	4.301	0.0434	−3.561	0.0023
CYR61	28.917	0.0000	−3.997	0.001	RIT1	17.399	0.0005	−2.506	0.0157
EHHADH	3.697	0.0000	−2.225	0.004	RPS6KB1	12.641	0.0001	−2.086	0.0472
EPHA2	3.509	0.0000	−2.418	0.049	SAV1	3.466	0.0036	−2.038	0.0209
ETS1	16.393	0.0000	−3.354	0.003	SENP2	9.278	0.0002	−2.756	0.0078
F3	5.165	0.0003	−3.409	0.003	SERPINE1	3.547	0.0016	−4.902	0.0018
HHAT	6.855	0.0334	−2.084	0.041	SGPL1	4.066	0.0002	−2.714	0.0106
KAT2B	20.055	0.0027	−2.428	0.045	SMPD1	2.1766	0.0011	−3.8267	0.0158
KCTD11	44.393	0.0009	−3.208	0.025	SMURF2	8.1538	0.0007	−2.7949	0.0174
M6PR	3.165	0.0008	−2.590	0.007	SRF	20.2462	0.0000	−2.5036	0.0222
MAP3K1	3.446	0.0061	−2.085	0.027	SSH2	4.7555	0.0001	−2.7527	0.0168
MAP3K14	2.762	0.0011	−2.830	0.020	SYN2	3.0399	0.0349	−2.7661	0.0085
MAP4K4	13.365	0.0001	−2.783	0.015	TBK1	4.9114	0.0003	−2.6957	0.0153
MAPKAPK2	3.004	0.0028	−2.989	0.017	TEAD4	14.2232	0.0001	−2.3833	0.0489
NRAS	4.051	0.0043	−3.585	0.002	TRAF5	7.1550	0.0042	−2.1078	0.0394
NUMB	2.974	0.0016	−2.286	0.041	UNC13D	2.6362	0.0043	−2.7414	0.0298
PDLIM2	6.149	0.0002	−2.971	0.034	YAP1	31.9441	0.0002	−2.9568	0.0125
PLCG1	25.086	0.0000	−2.500	0.031	YES1	6.5590	0.0002	−4.2868	0.0046
PPM1F	9.955	0.0000	−2.442	0.035	ZMYND15	7.6683	0.0002	−2.8244	0.0094

**Table 2 ijms-21-04514-t002:** List of mRNA selected by cDNA array analysis in SGPL1-downregulated cells.

Gene Symbol	Fold Change	Gene Symbol	Fold Change	Gene Symbol	Fold Change	Gene Symbol	Fold Change
8-Mar	−2.1	ERO1B	2.01	LSM12	−2.41	SDR16C5	2.41
ABCD3	−2.08	F8A2; F8A3; F8A1	2.24	LSM12	−2.32	SEC61B	−2.9
ABI2	−2	F8A3	2.19	LYN	−2.13	SERPINB9	−3.33
ADAM29	−2.4	FAM117B	2.5	MAZ	2.73	SERPINE1	−2.59
ADAMTS17	−2.62	FAM195A	2.25	MED22	−2.08	SGPL1	−9.74
AMMECR1	3.66	FAM49B	−3.32	MED28	2.09	SH3GL3	−2.35
AMY2B; ACTG1P4	2.03	FAS	2.57	MFAP3L	2.51	SHROOM3	−3.1
ANO6	−2.41	FBXL13	−2.55	MRS2	3.76	SLC19A2	3.27
AP1S3	−2.04	FEM1C	2.28	MTA1	2.02	SRPR	−3.45
AP3S2; MIR5009	−2.33	FGF19	−2.3	MYEOV	−2.02	ST14	−4.31
BHLHA9	−2.02	FOXM1	−2.92	MYO6	−2.33	STAU1	−2.23
BIRC3	−2.08	FYCO1	−2.26	MYO9B	2.31	STX8	−2.23
BLOC1S5-TXNDC5	−2.54	GABRA1	2.28	NACC2	2.31	TAB3	−2.11
boyboy; RP4-630A11.3; LEPR	−2.09	GABRQ	−2.16	NCOA7	−2.61	TAF5L	2.05
C14orf1	3.14	GBF1	−2.42	NFIX	2.43	TBL1X	−2.01
C1orf159	2.22	GMNN	−2.51	NLRP14	2.05	TCEAL2	−2.21
C1QTNF3	2.06	GPR183	−2.24	NMRK1	2.04	TEX264	2.5
CACNA2D3	−2.02	GRAMD4	2.29	NR4A2	2.06	TFRC	−2.85
CBX6	2.11	GRASP	−2.02	PABPC4L	−2.19	TGFA	−2.94
CCNG2	3.25	HDAC9	−2.26	PANK1	2.68	TGM2	−2.03
CD151	−2.29	HGD	−2.3	PARP8	−3.94	THBS1	−2.03
CD180	2.71	HIST1H1B	−2.26	PGAP1	2.07	THNSL2	−2.08
CDKN1A	3.04	HIST1H2AK	−2.07	PGM1	−2.34	TIMM8A	2.17
CFB	−2.01	HOXA7	−2.99	PIK3R2; IFI30	2.7	TJP1	3.2
CHST3	−2.27	IFNGR2	2.54	PKP4	−2.12	TMEM117	−2.46
CLASP2	−3.09	IQCK	−3.07	PLAGL2	−2.9	TMEM64	−3.94
CLDN1	2.11	ITPK1	−2.2	PNRC1	2.01	TMTC3	−2.02
CPNE9	−2.15	JAK2	−2.42	POGK	−2.02	TNFSF13B	2.1
CRLF1	2.14	JUN	−2.38	POLR2A	−2.5	TNRC6A	−2.04
CYB5R3	2.75	KCNAB3	−2.79	PON3	2.26	TRIB2	2.85
CYP2S1	−2.08	KCNK13	−2.14	PPIE	−2.07	TSNAX	−2.43
DCAKD	−2.49	KIR2DL4	−2.06	PPP1R2	2.07	TXNDC5	−2.6
DCLK1	−2.02	KLF6	−2.27	PRC1	2.41	UBE2D3	2.48
DEFB131	2.07	KLF7	−2.18	PRKAR1A	−3.94	UNC119B	2.2
DHRS13	2.76	KPNA1	−2.05	PTGFRN	−2.79	USP12	−2.2
DNAJC19	2.24	LAMTOR3	−3.27	RAB11FIP1	−2.98	USP2	−2.03
DNM3	2.26	LIMA1	−2.01	RAB30	−2.4	USP41	2.25
DOCK7	2.09	LINC00260	2.15	RAC2	−2.92	VOPP1	−2.07
DR1	−2.1	LMLN	−3.03	RAD21	2.02	VPS8	−2.2
DUSP18	−2.28	LOC100287225; RP11-267C16.1	−2.11	RGS22	2.52	WDR37	2.31
EGR1	−2.2	LRP12	−2.38	RHOB	−2.57		
EPN3	2.04	LRRIQ3	−2.12	S100A5	−2.02

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
