# Peer review of "Heterogenous Nuclear Ribonucleoprotein H1 Promotes Colorectal Cancer Progression through the Stabilization of mRNA of Sphingosine-1-Phosphate Lyase 1"

_ijms, 2020, doi:10.3390/ijms21124514_

Round 1

Reviewer 1 Report

This study investigated whether the mechanism underlying colorectal cancer cell progression mediated by hnRNP H1. The present study showed that hnRNP H1 promoted colorectal cancer cell HCT116 progression by inhibiting cell apoptosis, and that SGPL1 is involved in the hnRNP H1-mediated promotion of HCT116. In addition, the authors suggested that hnRNP H1-induced stabilization of SGPL1 mRNA promoted tumor progression by inhibiting p53 phosphorylation in HCT116 cells. I think that the topic of this manuscript and obtained results is interesting. However, there are several flaws in the manuscript. The most serious issue is that only one colorectal cancer cell line and only one siRNA targeting hnRNP H1 were used in this study. To validate the present results and to exclude a possibility that the obtained results were due to off-target effect of siRNA, the authors have to repeat some of the experiments using other colorectal cancer cell lines and other siRNA targeting hnRNP H1.

 Other comments

  1. Figure 1B and 1D: The authors have to confirm that the hnRNP H1 protein expression, not mRNA expression, was downregulated by transfection with siRNA targeting hnRNP H1. Similarly, the SGPL1 protein expression in the SGPL1 knockdown cells should be examined.
  2. Figure 1C, D: The source of HCEC-1CT is missing.
  3. Figure 2A: What database was used? Please add the information about it.
  4. Figure 2B: How did you get the specimens? In addition, the description about ethical approval is missing.
  5. Figure 4A: The source of primary antibodies is missing.
  6. Figure 4B: What kind of pifithrin was used? Pifithrin-α or Pifithrin-μ? In addition, the information about the treatment period or culture period is missing.
  7. Figure legends: The description about sample size is missing.
  8. Reat-time PCR: The sequences of primers used in this study is missing.    
  9. There are many careless mistakes. For example, “Figure 4” in Line 120-130 should be “Figure 3”. In addition, “siRNA of hnRNP H1” in Figure 2C and Figure 3B should be “hnRNP H1”. The authors have to check the manuscript carefully.
  10. I recommend that a native speaker of English reviews the manuscript to improve word choice, sentence structure, and grammar.

Reviewer 2 Report

The authors examined the mechanism of the tumor-promoting properties of the hnRNP H1 protein. They showed that the SGPL1 is an important effector of hnRNP H1 with respet to tumor growth.

  1. The HCEC-1CT cells and their cultivation is not described in the methods part.
  2. Figure1: in panels A and C, y-axis labels are missing (OD510?). It would be even better to show fold changes relative to scramble siRNA at 24h.
  3. Figure 1: the siRNA efficieny looks good at mRNA level. However it would be good to perform western blot analyses to show hnRNP H1 protein levels after knockdown.
  4. Figure 2: which public cohort was analyzed in Fig 2A (GSE number)?
  5. Figure 2: How many patients were analyzed in Fig 2B? Are normal tissue and tumors from the same patient (matched)?
  6. Figure 3E: instead of knockdown of hnRNP H1 and SGPL1 simultaneously, it would be better to perform a rescue experiment. Knockdown of hnRNP H1 and overexpression of SGPL1, to examine whether ectopic expression of SGPL1 rescues the hnRNP H1-knockdown mediated reduction of cell viability.
  7. Table 2: a bar chart would be better
  8. Methods: the ID of the hnRNP H1 antibody from Abcam should be provided
  9. Methods, binding assay: It is indicated that RNA isolation, reverse transcription kits, and qPCR was done with microRNA kits. However, the authors focused on mRNAs not on microRNAs.
  10. Methods Xenografts: it is indicated that BALB/c mice were used. Probably, nude mice is correct.
  11. All experiments were performed with HCT116 cells. The key results should be also performed using an additional cell line.
  12. The authors suggest that the repression of p53 is crucial for tumor-promoting properties of the hnRNP H1 protein. To verify this, a p53 mutant or p53 null cell line should be used and analyzed whether knockdown of hnRNP H1 has no effect in these cells. The p53 knockout HCT116 cells are available.
  13. The authors showed that the expression of hnRNP H1 is increased in CRC tumors. How about the levels of SGPL1 in tumors versus adjacent normal colon mucosa?

Round 2

Reviewer 1 Report

The authors have responded to all of my comments. I am satisfied with the authors’ response.

Reviewer 2 Report

The authors have properly addressed all concerns.